

# Will a perfect model agree with perfect observations? The impact of spatial sampling.

N.A.J. Schutgens[1], E. Gryspeerdt[2], N. Weigum[1], S. Tsyro[3], D. Goto[4], M. Schulz[3], and P. Stier[1]

[1]Department of Physics, University of Oxford, Parks road, OX1 3PU, England
[2]Institute for Meteorology, University of Leipzig, Stephanstr. 3, 04103 Leipzig, Germany
[3]Norwegian Meteorological Institute, O313 Oslo, Norway
[4]National Institute for Environmental Studies, 16-2 Onogawa, Tsukuba, 305-8568, Japan

*Correspondence to:* Nick Schutgens (schutgens@physics.ox.ac.uk)

**Abstract.** The spatial resolution of global climate models with interactive aerosol and the observations used to evaluate them is very different. Current models use grid-spacings of $\sim 200$ km, while satellite observations of aerosol use so-called pixels of $\sim 10$ km. Ground site or air-borne observations concern even smaller spatial scales. We study the errors incurred due to different resolutions by aggregating high-resolution simulations (10 km grid-spacing) over either the large areas of global model
grid-boxes ("perfect" model data) or small areas corresponding to the pixels of satellite measurements or the field-of-view of ground-sites ("perfect" observations). Our analysis suggests that instantaneous RMS differences between these perfect observations and perfect global models can easily amount to 30–160%, for a range of observables like AOT (Aerosol Optical Thickness), extinction, black carbon mass concentrations, $PM_{2.5}$, number densities and CCN (Cloud Condensation Nuclei). These differences, due entirely to different spatial sampling of models and observations, are often larger than measurement
errors in real observations. Temporal averaging over a month of data reduces these differences more strongly for some observables (e.g. a three-fold reduction i.c. AOT), than for others (e.g. a two-fold reduction for surface black carbon concentrations), but significant RMS differences remain (10-75%). Note that this study ignores the issue of temporal sampling of real observations, which is likely to affect our present monthly error estimates. We examine several other strategies (e.g. spatial aggregation of observations, interpolation of model data) for reducing these differences and show their effectiveness. Finally, we examine
consequences for the use of flight campaign data in global model evaluation and show that significant biases may be introduced depending on the flight strategy used.

## 1 Introduction

Airborne aerosols are a fascinating component of the Earth's atmosphere. They come in a bewildering variety of shapes, sizes and compositions. More importantly, they can affect the radiative budget and energy and hydrological balances of the
atmosphere (Angstrom, 1962; Twomey, 1974; Albrecht, 1989; Hansen et al., 1997; Lohmann and Feichter, 2005, 1997). Dust aerosols may transport nutrients for the biosphere over long distances (Swap et al., 1992; Vink and Measures, 2001; McTainsh and Strong, 2007; Maher et al., 2010; Lequy et al., 2012) and air pollution aerosol can pose health hazards for humans (Dockery et al., 1993; Brunekreef and Holgate, 2002; Ezzati et al., 2002; Smith et al., 2009; Beelen et al., 2013). Aerosols have also been suggested as disease vectors (Ballester et al., 2013). For a recent review of some of these aspects, see Fuzzi et al. (2015).





Models provide powerful tools to explore the role of aerosols, but require evaluation against observations in order to quantify their skill and detect possible model errors. AEROCOM is an international community of scientists (http://aerocom.met.no) involved in evaluating global aerosol models (Kinne et al., 2006; Schulz et al., 2006; Textor et al., 2006, 2007; Huneeus et al., 2011; Koch et al., 2009; Quaas et al., 2009; Koffi et al., 2012) but model evaluations are also routinely performed by individual research groups around the world. It is therefore surprising that evaluation strategies themselves have received relatively little scrutiny.

Due to constraints on computational resources, global aerosol-climate models are currently run at spatial resolutions of $\sim 200$ km. This of course limits their ability to resolve fine-scale structure (Benkovitz and Schwartz, 1997; Weigum et al., 2012) which will affect the comparison of global model data with observations: models and observations represent averages over different spatial areas. Satellite remote sensing observations are made for nominal pixels of 10 km as for MODIS (MODerate resolution Imaging Spectroradiometer) or 17 km as for MISR (Multi-angle Imaging SpectroRadiometer) or 3 km as for SEVIRI (Spinning Enhanced Visible and InfraRed Imager). Ground stations from AERONET can be estimated to sample no more than 5 km horizontally away from the site. In-situ measurements see even less of the atmosphere surrounding them. Yet, observed aerosol fields are known to exhibit variations over relatively short distances of 10 to 100 km (Anderson et al., 2003; Kovacs, 2006; Santese et al., 2007; Shinozuka and Redemann, 2011; Schutgens et al., 2013). Note that the spatial resolution of global models also impacts global model data due to the non-linear nature of many physical and chemical processes (Qian et al., 2010; Gustafson et al., 2011; Stroud et al., 2011; Weigum et al., 2015), but that is not the topic of this paper.

We use high-resolution model simulations (with a 10 km grid-spacing) to simulate both perfect global model data and perfect observations. These data are considered perfect in the sense that they are both derived from the same high-resolution simulation that we treat as the truth. In fact, the only difference between the global model data and observations is the area over which the high-resolution simulation is averaged (see Sect. 3). No measurement errors are added to the observations. The high-resolution simulations allow us to build up statistics of the difference between observations and model data, under a large variety of scenarios. In particular, we consider different observables like AOT, $PM_{2.5}$, number densities and CCN for different regions on the globe. We also evaluate a variety of averaging and interpolation strategies designed to bring model data and observations closer together. These high-resolution model simulations provide us with a toy model of what happens when global model data are evaluated with observations, ignoring both model and observation errors.

Since we simulate global model data as an average over the high-resolution data, a very relevant question is: 'what average is appropriate?'. This question is closely tied to the question of what the grid-point value of a global model represents and will be addressed later.

Section 2 introduces the three different models and 6 different regions for which we have high-resolution simulations. We also explain how the simulated fields were turned into observables. Section 3 describes in more detail how both global model data and observations are generated from the high-resolution simulations. In particular, Sect. 3.1 discusses various interpretations that may be given to a global model's grid-point value. Section 4 then introduces the concept of spatial sampling as a source of error through some examples. More substantive statistics can be found in Sect. 5, 6, 7, 8 and 9. An evaluation



of several strategies to reduce spatial sampling differences is given in Sect. 10. A preliminary analysis of the consequences of spatial sampling for the use of flight campaign data can be found in Sect. 11. The paper concludes with a summary (Sect. 12)

Note that Sect. 3.2 contains some general guidelines to interpreting many of the figures and statistics that appear in this paper.

## 2   The regional models

Three different regional models were used to create high-resolution simulated fields (10 km, 1 hour) of common observables: AOT, extinction, $PM_{2.5}$, black carbon mass concentration, number densities and CCN. Fig. 1 shows the simulation regions, and Table 1 summarises the most important information on these simulations.

WRF-Chem (Grell et al., 2005; Fast et al., 2006) was run for three regions using the MADE/SORGAM aerosol module (Ackermann et al., 1998; Schell et al., 2001), and one region using the GOCART bulk aerosol scheme. The meteorology was nudged to NCEP-FNL operational analysis data. The West-Europe and Oklahoma runs used emission scenarios (TNO MEGAPOLI-2005 or US National Emissions Inventory NEI-2005) with imposed 24-hour cycles for the anthropogenic emissions. These regions are characterised by fairly localised spatially-fixed sources. The Congo experiment used daily biomass burning emissions derived from MODIS fire counts and is characterised by highly localised sources that differ in location from day to day. The MADE/SORGAM module assumes the aerosol to exist in three modes (Aitken, accumulation and coarse) of varying species mixtures (sulfate, nitrate, organic and black carbon, sea salt and dust). MADE/SORGAM explicitly treats nitrates and SOA (secondary organic aerosol).

An expanded version of EMEP/MSC-W (Simpson et al., 2012) that includes calculations of aerosol bulk optical properties (based on work by Hess et al. (1998) and Chin et al. (2002)) was run at a $0.1^{\circ} \times 0.1^{\circ}$ grid, using ECMWF-IFS meteorology for 2008 and TNO-INERIS emissions for 2009 for Europe. Emissions of black carbon were derived from from the emissions of primary $PM_{2.5}$, using EMEP standard split-factors (per country and sector). Monthly, day-of-week and hourly temporal profiles were applied to the annual emissions. The EMEP chemical scheme includes approximately 160 reactions. The aerosols are represented as bulk-mass distributed between a fine fraction (including sulfate, nitrate, ammonium, organic and black carbon sea salt and dust) and a coarse fraction (nitrate, sea-salt and dust). Ammonium nitrate is calculated with the equilibrium model MARS, and the formation of coarse nitrate from nitric acid depends on the relative humidity. SOA is calculated using the VBS approach. For all details see Simpson et al. (2012) and references therein.

NICAM-SPRINTARS (see Goto et al. (2015) and references therein) was run in global mode with a stretched grid that had a resolution of 11 km over a part of Honshu (main island Japan). Its meteorology was nudged to NCEP-FNL reanalysis data. SPRINTARS uses a mass bulk scheme with individual modes for sulfate, organic carbon, black carbon and bins for sea-salt and dust. Two different organic/black carbon mixtures are also represented by individual modes. Anthropogenic emissions of black carbon and the SO4 precursor gas SO2 had a prescribed diurnal cycle. SOA were treated in the simple manner of scaling aerosol emissions. Nitrate aerosol were ignored in this SPRINTARS simulation.





Both EMEP and SPRINTARS do not calculate number densities as prognostic variables (SPRINTARS can diagnose them from assumed size distributions) and consequently did not provide those fields for our analysis. Both EMEP and SPRINTARS data were regridded from their original model grids to regular grids with 10 km spacings.

## 2.1 Observable parameters

The simulated fields examined in this paper are, for obvious reasons, all observables, see Table 2. All of the models provided AOT, extinction and (dry) $PM_{2.5}$, although WRF-Chem calculates AOT and extinction for 600 nm and EMEP and NICAM-SPRINTARS for 550 nm. WRF-Chem MADE also provided number densities and CCN at various super-saturations $S$.

   Real black carbon measurements by SP2 (single particle soot photometer) require a minimum black carbon content per particle. Due to redistribution of black carbon mass over many particles of mixed species, modal aerosol schemes cannot

properly simulate SP2 measurements, see also (Kipling et al., 2013). We decided to ignore this minimum black carbon content and used the total black carbon concentration as provided by the model.

   Number densities (N3, N10 and N50, i.e. number densities for particles with wet diameters in excess of 3, 10 or 50 nm) are constrained by a lower size threshold. In actual measurements, this lower size threshold applies to particles in relatively dry air but WRF-Chem provides only modal information for ambient humidities. However, based on dry $PM_{2.5}$ and its associated

water content data, we estimate that 'taking out' the water has at most a 10% effect on N10 or N50 values. Furthermore, from $PM_{2.5}$ data we also conclude that taking account of dry conditions may actually increase the spatial sampling errors we are studying.

   Finally, since WRF-Chem calculates the equilibrium of the ammonia & nitric-acid & sulfuric-acid & water system Seinfeld and Pandis (2006), 'drying out' the modelled particles amounts to much more than simply removing the water: it would lead to

a shift in the equilibrium. Currently WRF-Chem provides no mechanism to simulate this aspect of observed number densities.

## 3 Simulating observational and global model data

This section briefly describes the main methodology used in this paper. Using the high resolution simulated fields, we have generated both perfect observations and perfect global model data. The high resolution field $v$ has a regular rectilinear horizontal grid ($10 \times 10$ km), and a regular temporal spacing (1 hour). Only the vertical spacing is non regular and differs among the

models. The field $v$ can be thought of as 3 or 4-dimensional data cubes $v_{xyt}$ or $v_{xyzt}$ where $x = 1 \ldots n_x$ and $y = 1 \ldots n_y$ are indices to the horizontal coordinates, $z = 1 \ldots n_z$ is an index to the vertical coordinate and $t = 1 \ldots n_t$ is an index to the time coordinate. In the following, the $z$ coordinate is ignored for brevity's sake. A single perfect observation $O_{xyt}$ at time $t$ and location $x, y$ is simulated by:

$$O_{xyt} = v_{xyt}. \tag{1}$$



A perfect global model grid point's value $M_{xyt}$ can be simulated by averaging $v_{xyt}$ over a global model grid-box area $(2\Delta x + 1) \times (2\Delta y + 1)$ in the high-resolution field:

$$M_{xyt} = \sum_{i=-\Delta x}^{\Delta x} \sum_{j=-\Delta y}^{\Delta y} w_{ij} v_{x+i;y+j;t}, \tag{2}$$

where $\Delta x$ and $\Delta y$ represent the longitudinal and latitudinal half-sizes of a grid-box, as measured in the coordinate indices. Here $w$ is a normalised weighting function (to be defined later). Note that perfect model data can only be calculated on an inner domain of the high-resolution run of $1 + \Delta x \leq x \leq n_x - \Delta x; 1 + \Delta y \leq y \leq n_y - \Delta y$.

In the case that the location of the observation and the grid-point coincide, an instantaneous spatial sampling error can now be defined as:

$$\epsilon_{xyt} = O_{xyt} - M_{xyt} \tag{3}$$

where we use the perfect model value as a reference, since it is the model value that we want to evaluate in actual comparisons of observational and model data. It is straightforward to define a relative sampling error for time-averaged data by

$$\varepsilon_{xyt} = \left( \sum_{k=t-\Delta t}^{k=t+\Delta t} O_{xyt} - M_{xyt} \right) \Big/ \left( \sum_{k=t-\Delta t}^{k=t+\Delta t} M_{xyt} \right), \tag{4}$$

where $2\Delta t + 1$ is an arbitrary averaging interval.

A subset of the data cube of our regional simulations is used to build up error statistics. In addition to the limitation imposed by the Eq. 2 (already discussed), the outer 50 km of the simulated region was excluded from our analysis to reduce boundary effects. Similarly, the first two days of each simulation were used as a spin-up and excluded from analysis. At various points in our analysis, we have studied the sensitivity of our results to these choices but found no significant impact.

### 3.1 Interpretation of the grid-point value

We generate the global model grid-point value $M_{xyt}$ as a weighted average of the high-resolution simulation over a large area, see Eq. 2. The weighting function $w$ represents our interpretation of the global model's grid-point value. The question is what are realistic $w$ like for actual global models?

A numerical grid with spacing $\Delta$ can represent standing or travelling waves with a wavelength of in theory $2\Delta$ and in practice $4\Delta - 6\Delta$. This suggests that the grid-point value of a low resolution model is at best some average of a high resolution simulation over the grid-box $\Delta \times \Delta$. More-over, at horizontal resolutions of $\sim 200$ km, there is no evidence that actual global models have converged numerically (Pope and Stratton, 2002; Roeckner et al., 2006; Williamson, 2008). As the resolution of global models is increased, various aspects of the models are tweaked to obtain best agreement with either observations or reanalysis datasets (see Pope and Stratton (2002) for a very clear description). Diffusion is adapted to prevent numerical instabilities and the gravity-wave drag coefficients are modified according to the resolution of the orography. Most famously, various parameters related to sub-grid cloud processes are tuned to obtain radiative balance at the top-of-atmosphere. Our point here is that the strategy for tweaking the global model to best reflect an observational or reanalysis dataset effectively





determines $w$, although this is never explicitly discussed. In addition, models are tuned for only a few parameters for which abundant observations or reliable reanalysis data are available (e.g. pressure, temperature). There is no reason to assume that other parameters require the same weighting function, as these models are non-linear.

Hence we argue that $w$ is fundamentally unknown (and may actually vary with time and location). To conduct our analysis, we therefore assumed three different weighting functions and performed sensitivity studies (to be described later). The weighting function most used in this paper is a constant value throughout the grid-box. This corresponds to the mental model that many scientists have of the physics processes that occur in a grid-box. The other two weighting functions favour the area near the grid-point more than the outer edges of the grid-box. One weighting function uses a linear profile (highest at the grid-point, zero at the edge) and another uses a Dirac-$\delta$ (centred at the grid-point). The latter we consider a rather unlikely choice of $w$ but it does correspond to the case where the model has numerically converged.

## 3.2 Some conventions used in this paper

This paper contains many figures and statistics of spatial sampling error distributions. Instead of repeating the same information, some aspects are explained here. Error distributions are always given for either instantaneous ('hourly') or monthly data over a single region, see Table 1. These error distributions are quantified through Root-Mean-Square (RMS) values or quantiles. They represent typical errors per region (over no more than a month), which should not be mistaken for the typical error in any one longitude/latitude location. We use the so-called parametric 7-number summary of the 2, 9, 25, 75, 91 and 98% quantiles $q$ of the errors because for a normal distribution these quantiles are equally spaced. Any skewness or extended wings in a distribution will be readily visible. In particular, we often refer to the inter-quantile ranges $\Delta q_{50} = q_{75} - q_{25}$, $\Delta q_{82} = q_{91} - q_9$ and $\Delta q_{96} = q_{98} - q_2$. In e.g. Fig. 5 different shades of grey are used to denote these interquantile ranges: light grey for $\Delta q_{96}$, medium grey for the $\Delta q_{82}$ and dark grey for $\Delta q_{50}$. The solid blue line represents the median error. In Fig. 6, box-whisker plots show the error distributions. Different widths of the bars are used to denote different inter-quantile ranges: narrow for $\Delta q_{96}$, medium for $\Delta q_{82}$ and wide for $\Delta q_{50}$. The black rectangle represents the median error and the black circle the mean error. In a few figures, additional error distributions are shown using colored lines: the $\Delta q_{50}$, $\Delta q_{82}$ and $\Delta q_{96}$ ranges will be indicated by resp. solid, dashed and dotted lines.

The standard measure of uncertainty, the standard deviation, is of course half the $q_{84.1} - q_{15.9}$ inter-quantile range. For a Gaussian distribution, $\Delta q_{50}$ is 1.35 times the standard deviation, and $\Delta q_{82}$ is 2.68 times the standard deviation. For a Gaussian distribution with zero mean, the standard deviation and the RMS value will of-course agree.

## 4 Examples of spatial sampling errors

In Fig. 2, we show instantaneous simulated AOT and surface black carbon concentration after 10 days in the WRF-Chem W-Europe run. By comparing the field in a small $10 \times 10$ km box to the average of a large $210 \times 210$ km box surrounding it (approximate size of present-day global model grid box), we assess spatial sampling errors. The centre of the large box we refer to as grid-point (of the global model). By moving these two boxes together throughout the region, we can build up statistics





of spatial sampling errors (also shown in Fig. 2). These errors can reach $\sim 100\%$ and form coherent patterns several global model grid-boxes large. Time series of the global model and observed values at a single location are shown in Fig. 3. In the case of AOT, we see that the perfect observation can both over- and under-estimate the perfect model value with variations on a time-scale of a day or so. The black carbon time-series, on the other hand, shows systematic underestimation by the perfect

observation over long periods for most of the month (note that events of overestimation also occur but on smaller time-scales). Although these time-series vary a lot throughout the region, this example is nevertheless typical.

Since these spatial sampling errors are substantial, it makes sense to try and reduce them by temporally averaging the data. In Fig. 4, we show monthly averaged simulated AOT and surface black carbon concentration from the same run. The spatial sampling errors in monthly averaged observations are also shown in Fig. 4. They are smaller than the errors for instantaneous

fields but still quite substantial (up to $\sim 20\%$ for AOT and $\sim 65\%$ for black carbon). Note also that the error patterns have become larger and more coherent. The effectiveness of temporal averaging is shown in Fig. 5, where the spatial sampling errors are shown as a function of averaging period. Time-averaging does decrease spatial sampling errors but not as fast as one would expect if instantaneous sampling errors were independent Gaussian noise. This is understandable because the persistence of emission sources and flow patterns in the atmosphere create temporal correlations in the fields of a few hours to a few days.

Note that AOT is more strongly (beneficially) affected by time-averaging than surface black carbon concentrations.

## 5   Agreement among models

Before studying these spatial sampling errors in more detail, we consider how (dis)similar they are among different models. The Europe region simulated by EMEP encompasses the W-Europe region simulated by WRF-Chem MADE and so these two models allow for ready intercomparison, see Fig. 6. We see that both instantaneous and monthly errors as predicted by

WRF-Chem and EMEP are of similar magnitude although WRF-Chem generally produces larger errors (note the exception of instantaneous errors for extinction near 2 km AGL). Error magnitudes for different observables behave similarly among WRF-Chem and EMEP: monthly errors for AOT and surface black carbon are the smallest resp. largest errors. EMEP monthly error maps (see Fig. 7) also look similar to WRF-Chem results (Fig. 4), especially for black carbon surface concentrations.

It would be interesting to understand the reason for the differences. From global model studies in the context of AEROCOM

(e.g. Myhre et al. (2013); Randles et al. (2013); Stier et al. (2013)), we know that such attribution is difficult and here we limit ourselves to pointing out some obvious differences between WRF-Chem and EMEP. First, there are differences in emission inventories and sea-salt emission parametrisations. Second, although both models were nudged to reanalysis data, transport will be different due to different dynamical cores and vertical resolution (WRF-Chem uses twice the vertical resolution as EMEP). For similar reasons wet and dry and wet deposition are different. Both models also use a very different aerosol scheme (mass

bulk or two moment scheme). All of this will affect aerosol life-times, which in turn will affect the spatio-temporal variability of aerosol.

It should also be pointed out that EMEP shows quite bit of month-to-month variation: e.g. January 2008 errors for AOT and March 2008 errors for surface black carbon concentration are markedly bigger than those estimated for May (not shown).



The most important point here is that both models suggest spatial sampling errors of similar magnitude with similar spatial patterns.

## 6   Different observables and different regions

Figure 8 shows relative spatial sampling errors (either instantaneous or monthly) for all observables and the three WRF-Chem MADE regions (see also Tab. 1 and Fig. 1). Instantaneous RMS errors are large: from 20 % up to 160% depending on observable and region (the RMS errors are calculated over a single region for the full month, see Table 1). There are clear and (mostly) systematic differences among the three regions in that W-Europe shows the largest errors and Congo the smallest. This may be related to the overall wind-flow: Congo shows the most laminar flow (and hence most coherent aerosol plumes), while W-Europe shows a very turbulent flow (we do not wish to discount other effects like the spatio-temporal distribution of sources but a full explanation is outside this paper's scope). Two observables (black carbon concentrations near 2 km AGL for all three regions and surface CCN at $S = 0.02\%$ in W-Europe) show errors down to -100%. In the case of black carbon, this is due to narrow black carbon plumes travelling through an otherwise pristine air layer: the observation often sees the pristine air but the model always includes contributions from the plume. In the case of CCN, the background CCN at $S = 0.02\%$ is very low, especially close to sources where many small particles are emitted. But once in a while a plume of larger particles travels over giving rise to much larger CCN at low super-saturation $S = 0.02\%$.

The monthly errors can be reduced quite a bit compared to the instantaneous errors. For many observables, RMS errors are 5–15%, although for e.g. observables like surface black carbon concentrations and N10 it can be resp. 30–50% and 30–80%, with individual errors reaching over 100%. Congo represents quite a different situation from the other two regions: the reduction due to averaging is much less, and in the case of surface N10 there is actually a slight increase in errors. An important difference between W-Europe & Oklahoma on one hand and Congo on the other is that the first have mostly fixed aerosol sources with a prescribed diurnal cycle. The latter has emission sources (fires) in different locations from day to day. The explanation for the large N10 errors is similar to that of the large instantaneous errors for black carbon plumes, except now the temporal extent should be taken into account as well.

Figure 9 shows relative spatial sampling errors for the other 3 regions, all simulated by models with mass-bulk schemes for aerosol. In general, spatial sampling errors appear to be a bit smaller than in Fig. 8, but note the exception of extinction near 2km AGL. Most monthly sampling errors over ocean are very low, presumably due to the short life-time and diffuse source regions of sea-salt aerosol. But large errors are found for extinction over ocean near 2 km AGL, that seem partly due to isolated plumes of sea-salt but mostly due to a broken cloud field that rains out sea-salt locally. Both instantaneous and monthly errors over Japan become larger if only observations over the land area are considered. The Japan region includes parts of the Japan sea and the North Pacific ocean that account for more than 80% of the simulated area. Also, the Japan simulation, like the Congo simulation, shows rather laminar flow from meso-scale to synoptic scale. Finally, simple statistics like in Fig. 9 cannot convey that over an extended region like Europe there are areas with systematically small or large sampling errors due to source locations and orography (see also Fig. 4 and 7).





In the case of actual observations, there may be quite a bit of intermittency in their temporal sampling suggesting that the spatial sampling decreases we have shown here for monthly averages represent a best case scenario.

## 7 Vertical distribution of sampling errors

The vertical distribution of spatial sampling errors can be very different depending on observable and region. Figures 10 and 11
show the instantaneous and monthly relative spatial sampling error profiles for extinction, N10 and black carbon concentrations.

We see that although errors are typically largest at and near the surface, this does not preclude large errors higher up in the atmosphere. The instantaneous errors for black carbon concentrations actually show largest errors from 2 to 7 km AGL. This is due to black carbon plumes in a relatively pristine background, which also explains why the error distribution is so clear skewed to negative values (observation sees the pristine background while the model also includes plumes). Black carbon's only source
is surface emission, but both extinction and N10 also have sources throughout the troposphere (nucleation, condensation and in-cloud production of sulfate) which likely explains the difference between these observables. Note that by design, WRF-Chem black carbon concentrations cannot go below a minimum value of $2 \cdot 10^{-16}$ $\mu$g/m$^3$; this should result in an underestimation of the sampling errors.

For the monthly errors, both extinction and black carbon concentrations show secondary maxima in sampling errors well
above the surface, while N10 errors drop steadily with altitude.

We have analysed the sampling errors at their original model levels, which for these simulations occur at fairly constant altitude above ground. Note that the errors estimated in this subsection do not take into account that a global model's grid-box may have a vertical extent larger than that of our regional simulations. Taking this into account would only increase the estimated errors. The profiles of spatial sampling errors for the bulk mass simulations are rather constant and therefore not
discussed here.

## 8 Impact of grid-box size and shape

### 8.1 Impact of latitude

Although our high resolution simulations were made at different latitudes on Earth, so far we have assumed that the global model grid-box size is equal to the grid-box size of a T63 grid at the equator (210 by 210 km). At higher latitudes, the
25 longitudinal extent of the grid-box shrinks (at least for rectangular grids), which may reduce spatial sampling errors. This is explored in Fig. 12. As we can see, smaller longitudinal extent leads to smaller errors although the effect is rather mild. When the longitudinal extent is halved, errors in monthly-averaged fields decrease between 10 and 30% of the original errors, with $\sim 20\%$ a very typical value. Also, larger errors are usually less affected than smaller errors although the differences are not very big. The figure for black carbon is typical, while AOT is rather the exception to this. Spatial sampling errors in
instantaneous fields behave very similar (not shown), although fields that show very large errors (like surface BC or surface CCN at $S = 0.02\%$) tend to show less improvement ($\sim 10\%$) when the grid-box longitudinal extent is halved.





Note that the longitudinal extent only has an impact on spatial sampling errors because there are spatial and temporal correlations in the aerosol fields. If these fields were independent random noise, decreasing longitudinal extent would barely have an impact on sampling errors.

### 8.2 Impact of grid-box size

The impact of model resolution is also easily explored, see Fig. 13. Aggregation errors decrease by 10 to 50% from T63 (210 by 210 km) to T106 (125 by 125 km, a third of the T63 grid-box area), with 40% a rather typical value. Surface observations are less affected with decreases of $\sim 30\%$, especially N10 whose spatial sampling errors in all three simulations only decreased by $\sim 20\%$ when the grid-box size was halved. For instantaneous values (not shown), the typical reduction in sampling error is smaller, $\sim 30\%$, especially for surface fields: $\sim 20\%$.

As with the longitudinal extent, gridbox-size only has an impact because of the spatial and temporal correlations in the aerosol fields. A field of independent random noise exhibits sampling errors quite independently of gridbox-size (unless the box, and the number of values therein, becomes very small).

### 9 Observations offset from the grid point

Sofar we have considered observations at the exact grid-point of a global model's grid-box which is a useful starting point but also quite unrealistic. For a sample of randomly distributed observations in a 210 by 210 km grid-box, only 2% will be within 10 km of the grid-point and 50% will be more than 84 km away from it. By considering observations located throughout the grid-box, and not just its centre, it is possible to show how monthly sampling errors increase with distance of the observation to the grid-point, see Fig. 14. As a matter of fact, 50% of possible AOT observations have errors at least twice as large as found for observation coinciding with the grid-point. Observations in the very corners of the grid-box exhibit errors three times as large.The increase of sampling errors with distance to the grid-point for surface black carbon concentrations is not as large but still significant. Figure 15 shows box-whisker plots of monthly sampling errors for several observables, either at the grid-point, or at a distance of 70 or 100 km, for the W-Europe region. Similar results can be shown for Oklahoma and Congo, where the relative increase with distance is often (but not always) larger. For all three regions and all observables, the increase for $\Delta q_{82}$ at 70 km is between $1.2 - 2.3\times$ and the increase at 100 km is between $1.4 - 3.4\times$. Instantaneous spatial sampling errors increase less fast with distance but still significantly: typical increases for $\Delta q_{82}$ at 70 km is 1.3 for AOT and 1.2 for surface black carbon concentration (i.e. monthly averaging is more beneficial for an observation at the grid-point than one at 70 km distance).

As discussed before (Sect. 3.1), the meaning of a global model's grid-point value is not obvious. Sofar we have assumed that the grid-point value is the unweighted average of the high-resolution field over the global model's grid-box (i.e. a constant weighting function $w$). Here, we explore how the sampling errors depend on different weighting functions. Fig. 16 shows how a constant, linear or Dirac-$\delta$ weighting function affects sampling errors as a function of distance to the grid-point. For the Dirac-$\delta$ weighting function, sampling errors are equal to zero at a distance of zero: the global model's value is equal to the observation (since both are perfect). But as distance increases, so will the spatial sampling errors. Actually, for distances larger





than $\sim 30$ km, the three very different weighting functions give rather similar sampling errors (but notice that more localised weighting functions yield larger errors as expected). Since for randomly distributed observations, only $\sim 6\%$ would be closer than 30 km to the grid-point, we feel it is justified to conclude that the shape of the weighting function has only a small impact on statistics of spatial sampling errors. The spatio-temporal variation of the field is far more important.

## 10 Strategies for reducing sampling errors

The typical sampling errors when the observation is at the model grid-point are lower than those for an observation offset from the grid-point. It seems unlikely that we can devise strategies to reduce "centre-of-grid-box" errors, other than temporal averaging (see Sect. 6) or further averaging global model data (and their associated observations) over multiple grid-boxes. But the sampling errors for observations offset from a grid-point might be reduced by proper screening, interpolation within the model grid, or considering multiple observations at the same time.

### 10.1 Observations close to the model grid-point

As Fig. 14 shows, the smallest spatial sampling errors occur for observations close to the model grid-point. As a matter of fact, within a distance of 30 km, there is hardly any change in the errors (note: this figure uses the constant weighting function). To keep sampling errors as small as possible, one might only select observations that are within 30 km of a model grid-point. For a T63 grid-box at the equator ($210 \times 210 = 44100$ km$^2$), that implies only 6% of randomly distributed observations would be usable, a substantial reduction of potential observational data. For an upper distance of 50 km, this increases to 18% of observations, still representing a significant loss of observational data.

One benefit of selecting only observations close to the grid-point is that here the impact of the weighting function is most pronounced (see also Fig. 16). So within 30 km of the grid-point, spatial sampling errors may actually be very small if the weighting function is highly localised. Since it is impossible to know the actual weighting function, it may be difficult to assess whether it is localised or not.

### 10.2 Aggregating observations over the model grid-box

It has been suggested (e.g. Sayer et al. (2010)) that aggregating observations over a model grid-box is the best strategy for comparing models with observations. Obviously, such a strategy is only possible for satellite data that provide contiguous wide swath observations (e.g. MODIS, MISR, POLDER, SEVIRI). More-over, it can be expected that the success of this strategy depends on the weighting function that is applicable. Figure 17 shows relative spatial sampling errors in case of observations that are spatially aggregated before comparison to the model (it is assumed the aggregation is space-filling). Here the model grid-point and the centre of the aggregated observations coincide. As a result, sampling errors go to zero for the constant weighting function as the observational aggregation approaches the extent of the grid-box. For the linear weighting function, we see that errors initially become smaller as the aggregation increases and then grow again as the observational aggregation





approaches the extent of the grid-box. Still, sampling errors are halved when aggregating observations over the full grid-box so there is clearly a benefit. The extreme weighting function of the Dirac-$\delta$ obviously leads to large errors.

For actual satellite measurements it will be difficult to observe the complete grid-box, due to e.g. cloud cover, sun glint or high surface albedo. Sayer et al. (2010) show that in the case of AATSR observations (nominal $10 \times 10$ km pixel) and the

GEOS-Chem model ($5^\circ \times 4^\circ$ grid-box) it is extremely unlikely that more than 50% of a model grid-box would be covered by observations, that is: space-filling aggregations over global model grid-boxes are very unlikely.

### 10.3 Multiple observations in a model grid-box

Instead of a space-filling aggregation, one could average multiple observations in the same grid-box before comparison to the grid-point value and hopefully reduce sampling errors. The idea here is that if the observations are sufficiently far apart and rep-

resent fairly independent samplings of the field within the grid-box, their average should be distributed closer to the (weighted) grid-box average than an individual observation. This is similar to the previous sub-section, except far fewer observations are needed and no space-filling aggregation is required. This strategy may be employed for surface sites as well as for satellite data.

Figure 18 show errors in case of 4 independently distributed observations throughout the grid-box. Clearly, averaging mul-

tiple observations helps to reduce spatial sampling errors, even when the Dirac-$\delta$ weighting function is assumed! But note that this improvement is less in case of a more localised weighting functions. For the constant weighting function, we also see that smallest errors now occur not at a distance of 0 km, but at a distance of 50 to 70 km (for the linear weighting function this minimum shifts closer to the grid-point). This is quite understandable: close to the grid-point multiple observations are clustered together. Hence they will not be very different. As distance increases, the randomly distributed observations sample

more of the grid-box. Obviously, using more observations than 4 will give better results (not shown).

Note that Fig. 18 does *not* suggest that *any* set of 4 observations reduces sampling errors: if those observations are very close together, averaging them will hardly improve on the error.

### 10.4 Interpolating model data among grid-points

By interpolating the model data to the location of an observation, it may be possible to reduce spatial sampling errors for

observations located away from the model grid-point. The idea is to construct virtual model data for a virtual grid-box centred on the observation. This interpolation can be performed in different ways; here we consider linear interpolation and distance-weighted averaging. Figure 19 shows that linear interpolation i.c. of a constant weighting function clearly has a beneficial effect on spatial sampling errors, especially for observations far from the global model's grid-point. Notice that from about 80 km distance, errors become constant and no longer increase with distance (they are always larger than the errors for an

observation *at* the grid-point). Obviously, the impact depends on weighting function and interpolation method, as shown in Fig. 20. Figure 20 shows that interpolation is most beneficial for observations farthest from the grid-point and can actually lead to larger errors close to the grid-point. Interestingly, the more localised the weighting function, the more beneficial the interpolation (presumably because the global model data are now identical to observations at the grid-point). Finally, this figure



shows that linear interpolation performs better than distance-weighted average. This holds for all observables and all regions we considered.

Much the same conclusions can be stated for instantaneous values, except that the beneficial impact of interpolation is less pronounced.

## 11  Flight campaigns

Unlike satellite or ground-site observations, measurements taken during a flight campaign cannot be properly averaged over time (at least on time-scales from days to months and longer). To simulate the (nearly) instantaneous measurements during horizontal legs of flight campaigns, we use narrow tracks: 10 km wide and 210 km long, centred on the grid-point and running in either East-West or North-South direction. Profiles of spatial sampling errors for such flight campaign data can be seen in Fig. 21. Compared to instantaneous point observations (also shown), the flight campaign observations are less affected by spatial sampling issues because they sample a larger part of the grid-box. Even so, significant instantaneous RMS errors exist, varying between 10-41% for extinction, 10-46% for N10 and 21-100% for black carbon concentrations at different altitudes and for different regions (these errors are for a best case scenario: a grid-box long flight path centred on the grid-point). For Congo, spatial sampling errors can be quite different depending on whether the flight path runs North-South or East-West Around 6 km AGL prevailing wind flows are East-West, resulting in similarly oriented plumes. If the flight track observations are within and along such a plume, spatial sampling errors will be large and positively biased. If the flight track observations are across such a plume, errors will be smaller and (over a large domain) unbiased.

The Congo results highlight a particular issue with flight campaign data: if the flight tracks deliberately have been chosen to follow observed aerosol plumes, perfect observations will overestimate perfect model values by significant amounts.

Near vertical legs of flight campaigns should experience errors like those discussed for point observations, Sec. 7, but notice that we do no consider the vertical extent of a global model's grid-box in our analysis.

## 12  Conclusions

The spatial resolutions of current global aerosol models and the observations used to evaluate them are very different. Model grid-point values are representative of areas of $\sim 200 \times 200$ km$^2$ but individual observations seldom see more than $\sim 10 \times 10$ km$^2$ of the atmosphere. This difference in 'field-of-view' should affect the evaluation of models with observations but has received little attention in the literature. We believe our paper is the first systematic and qualitative study of the differences between a perfect model and perfect observations due to spatial sampling.

Using high-resolution simulations for 6 different regions by 2 different regional models and 1 global model, we show that spatial sampling errors can be substantial across a range of observables (AOT, extinction, PM$_{2.5}$, black carbon concentrations, number concentrations and CCN). These spatial sampling errors fluctuate in time and space, depending on emission sources,





grid locations, weather and aerosol processes. Ultimately, they constitute a noise that will be present in any model evaluation and that can not be eliminated entirely unless model grid sizes become smaller than observational fields-of-view.

Assuming observations that do not coincide with the global model's grid-point but are offset by 80 km (54% of randomly located observations in a $210 \times 210$ km grid-box will be further away), the following statistics are offered. For instantaneous data, RMS spatial sampling errors are larger than 30%, typically between 40 and 80% and may go up to 160% (depending on observable and region). These errors are typically positively skewed and highly non-Gaussian. For monthly data, RMS sampling errors are larger than 10%, typically between 10 and 40% and may go up to 75% (depending on observable and region).

This noise can however be reduced: we have explored the impact of spatial or temporal averaging of data as well as selection of observations based on distance to a grid-point or interpolation of model data to the location of an observation. Our study suggests that while increased model resolution will of course be beneficial, resolutions will need to be 4 times higher ($50 \times 50$ km$^2$ grid-box area) before spatial sampling errors become significantly smaller. In the mean time, we recommend that both model data and observations are spatio-temporally averaged to ensure best agreement. Here the model data must first be spatially interpolated to and temporally collocated with the observation. Optimal averaging procedures will depend on the spatio-temporal sampling of the observations, the characteristics of the observable and the requirements of the scientific community, so we offer no single prescription although the results in this paper provide some guidelines. Optimal strategies for evaluating models with observations need to receive more attention from researchers.

Our results suggest that caution is needed when using in-situ measurements in global model evaluation. These measurements consistently led to larger spatial sampling errors than remote sensing measurements like AOT. For instance, monthly surface black carbon concentrations & number densities for our simulations have RMS spatial sampling errors of at least 30% and and up to 80%. Best case scenarios for flight campaign data still allowed spatial sampling errors of 100% and typically the observation would underestimate the model.

Regarding the large sampling errors in case of black carbon, other species (e.g. sulfate, sea-salt) were not explicitly analysed in this paper but show different results (not shown). Sulfate errors tend to be rather small, probably due to the multitude of sources and relatively long-life times. Sea-salt, on the other hand, shows large and systematic monthly sampling errors along coast lines (unsurprisingly). Given the size of our global model's grid-box, these errors extend quite far into land or over sea. The important point here is that sampling errors for species mass concentrations can be very different dependent on species and hence have a big impact on the evaluation of a model's particle speciation.

It is likely that the spatial sampling errors estimated in this paper are under-estimates. First, Qian et al. (2010) showed that model spatial variability over 75 km increased significantly (by 60 to 100%) when model resolution changed from 15 to 3 km. Our current high-resolution simulations have resolutions of 10 km. Second, our high-resolution simulations do not resolve fine-structure below 10 km while many in-situ measurements actually have fields-of-view on the order of millimetres to centimetres (e.g. particle inlets). Third, our models are more limited in the spatio-temporal variation of their emission sources than reality due to constant diurnal patterns. Finally, even high-resolution models will have to take a broad view of aerosol and describe average properties (e.g. mass and/or number densities) instead of modelling individual aerosols in all their variety. On the other





hand, it is possible that in areas far away from sources (e.g. the free troposphere over the remote ocean) aerosol has mixed sufficiently to strongly reduce these spatial sampling errors (e.g. HIPPO measurements over the Pacific, see also Weigum et al. (2012)). Our simulations do not really allow us to explore this scenario.

In the interest of comparing likes to likes, this paper does not consider the fact that real observations may have very intermit-

tent temporal sampling. Nor does it consider the impact that precipitation may have on spatio-temporal variability of aerosol (Gryspeerdt et al., 2015, for example). These issues are the subject of further investigation.

*Acknowledgements.* This work was supported by the Natural Environmental Research Council grant nr NE/J024252/1 (Global Aerosol Synthesis And Science Project).

E. Gryspeerdt acknowledges funding from the European Research Council under the EU Seventh Framework Programme FP7-306284

('QUAERERE'). D. Goto was supported by the Global Environment Research Fund S-12 of the Ministry of the Environment (MOE)/Japan, the Grant-in-Aid for Young Scientist B (Grant Number 26740010) of the Ministry of Education, Culture, Sports and Science and Technology (MEXT)/Japan, and KAKENHI/Innovative Areas (Grant Number 24110002) of MEXT/Japan. P. Stier acknowledges funding from the European Research Council under the EU Seventh Framework Programme (FP7/2007-2013) / ERC grant agreement FP7-280025.

Michael Schulz and Svetlana Tsyro acknowledge funding from the Norwegian Research Council under the KLIMAFORSK project

'AeroCom-P3'. Their work was supported by EMEP under UNECE.

The figures in this paper were prepared using David W. Fanning's coyote library for IDL.



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





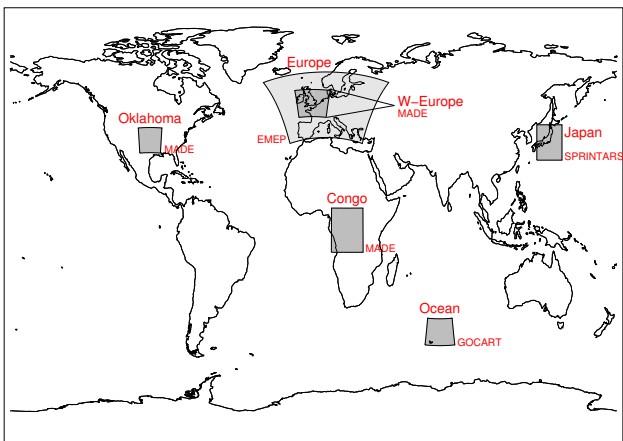

**Figure 1.** Three models were used in this study to simulate a variety of aerosol fields. The regional names used to identify these simulations are given in large font, while the models are denoted in small font. MADE and GOCART refer to the WRF-Chem version used.

within AeroCom, Atmospheric Chemistry and Physics, 6, 1777–1813, doi:10.5194/acp-6-1777-2006, http://www.atmos-chem-phys.net/6/1777/2006/, 2006.

Textor, C., Schulz, M., Guibert, S., Kinne, S., Balkanski, Y., Bauer, S., Berntsen, T., Berglen, T., Boucher, O., Chin, M., Dentener, F., Diehl, T., Feichter, J., Fillmore, D., Ginoux, P., Gong, S., Grini, A., Hendricks, J., Horowitz, L., Huang, P., Isaksen, I. S. a., Iversen, T., Kloster, S., Koch, D., Kirkevåg, A., Kristjansson, J. E., Krol, M., Lauer, A., Lamarque, J. F., Liu, X., Montanaro, V., Myhre, G., Penner, J. E., Pitari, G., Reddy, M. S., Seland, Ø., Stier, P., Takemura, T., and Tie, X.: The effect of harmonized emissions on aerosol properties in global models – an AeroCom experiment, Atmospheric Chemistry and Physics, 7, 4489–4501, doi:10.5194/acp-7-4489-2007, http://www.atmos-chem-phys.net/7/4489/2007/, 2007.

Twomey, S.: Pollution and the planetary albedo, Atmospheric Environment, 8, 1251–1256, 1974.

Vink, S. and Measures, C.: The role of dust deposition in determining surface water distributions of Al and Fe in the South West Atlantic, Deep Sea Research Part II, 48, 2787–2809, doi:10.1016/S0967-0645(01)00018-2, http://linkinghub.elsevier.com/retrieve/pii/S0967064501000182, 2001.

Weigum, N., Stier, P., and Schutgens, N.: Modelling of stratocumulus cloud layers in a large eddy simulation model with explicit microphysics, Atmospheric Chemistry and Physics, in preparation, 2015.

Weigum, N. M., Stier, P., Schwarz, J. P., Fahey, D. W., and Spackman, J. R.: Scales of variability of black carbon plumes over the Pacific Ocean, Geophysical Research Letters, 39, doi:10.1029/2012GL052127, http://doi.wiley.com/10.1029/2012GL052127, 2012.

Williamson, D. L.: Convergence of aqua-planet simulations with increasing resolution in the Community Atmospheric Model, Version 3, Tellus A, 60, 848–862, doi:10.1111/j.1600-0870.2008.00339.x, http://tellusa.net/index.php/tellusa/article/view/15499, 2008.





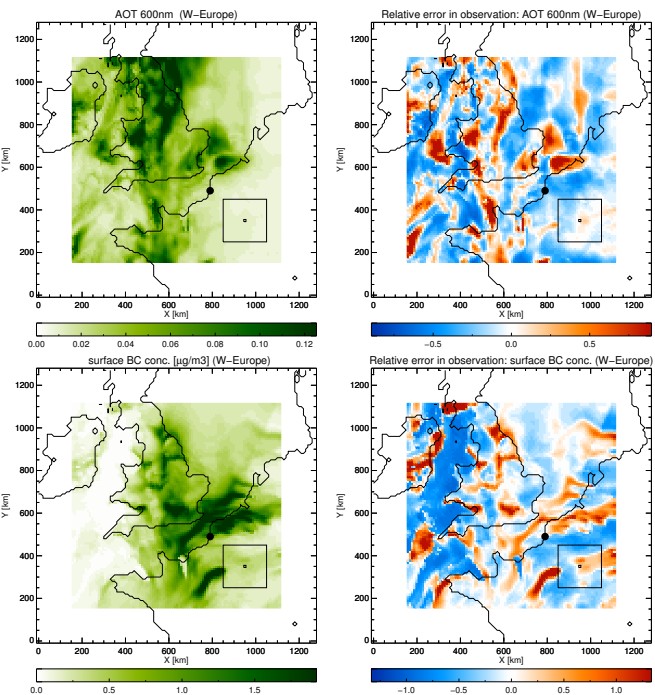

**Figure 2.** Snapshots of the simulated field and the relative spatial sampling error in the observation of AOT and surface black carbon concentration, as simulated over W-Europe at 10 days, 00 hours by WRF-Chem MADE. Also shown are two square boxes ($10 \times 10$ and $210 \times 210$ km) and a single location (fat dot), south of Calais, France. Note that the high-resolution simulations encompass the whole region shown, while our analysis is only made for the colored domain.





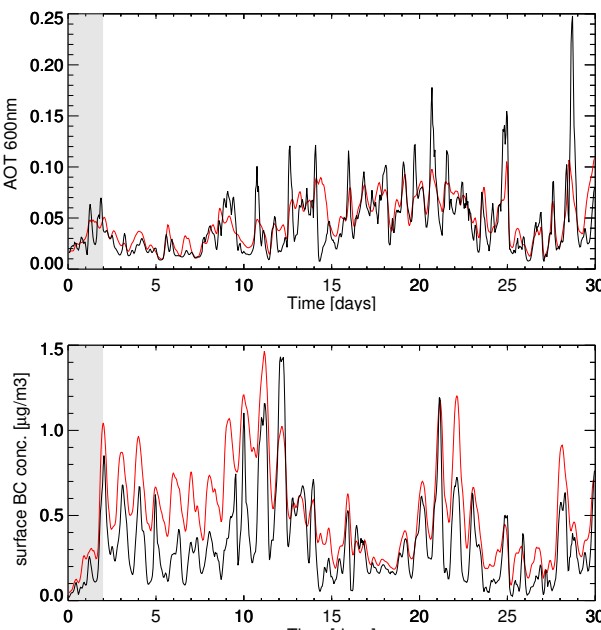

**Figure 3.** Timeseries of global model (red) and observed (black) AOT and surface black carbon concentration as simulated at a location south of Calais (France) by WRF-Chem MADE, see also Fig. 2. The grey bar to the left shows the model's spin-up period.





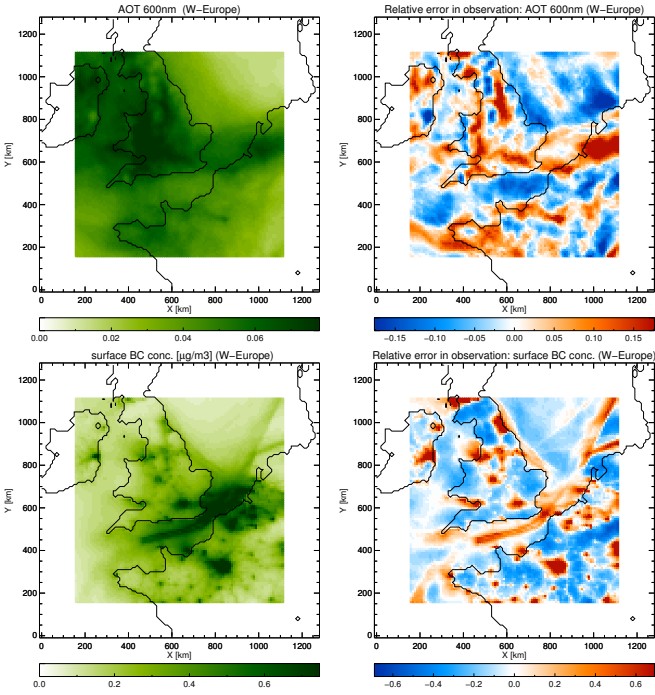

**Figure 4.** Monthly average of the simulated field and the relative spatial sampling error in the observation of AOT and surface black carbon concentration, as simulated over W-Europe by WRF-Chem MADE. Note that the high-resolution simulations encompass the whole region shown, while our analysis is only made for the colored domain.

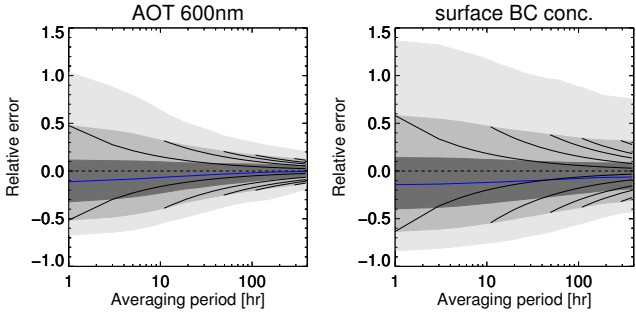

**Figure 5.** Relative spatial sampling error as a function of averaging period. The thin black lines are prognosis of the 9 and 91% quantiles *in case* these errors behaved like independent Gaussian errors (i.e. $1/\sqrt{n}$, with $n$ the number of observations). Results from WRF-Chem MADE over W-Europe. Further explanation in Sec. 3.2.





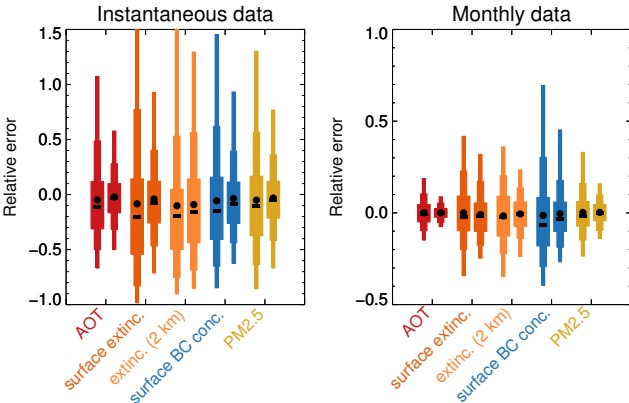

**Figure 6.** Relative spatial sampling errors (for either instantaneous or monthly data, note the different vertical axes) over the W-Europe region as calculated by WRF-Chem MADE (left bar) and EMEP (right bar) in May 2008. Further explanation in Sec. 3.2.

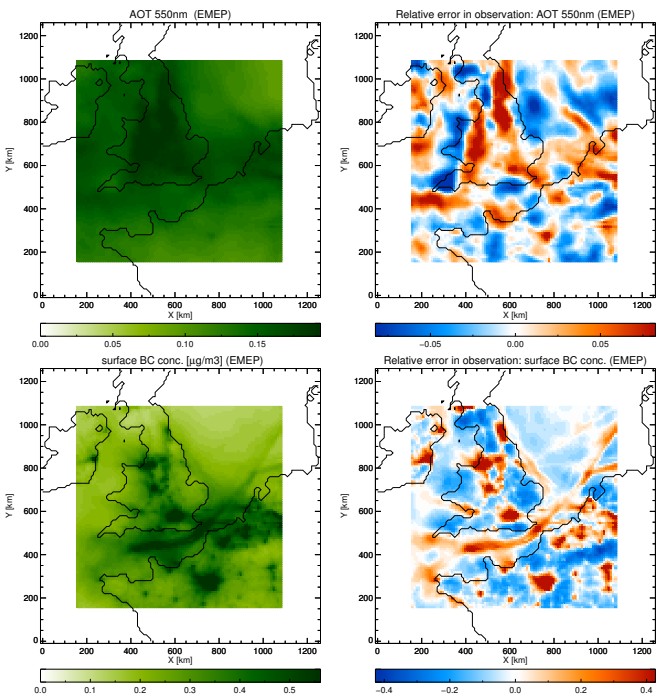

**Figure 7.** Monthly average of the simulated field and the relative spatial sampling error in the observation of AOT and surface black carbon concentration, as simulated over W-Europe by EMEP. This can be compared to results for WRF-Chem MADE as shown in Fig. 4 but note that the colour bars have different ranges to bring out spatial patterns better.



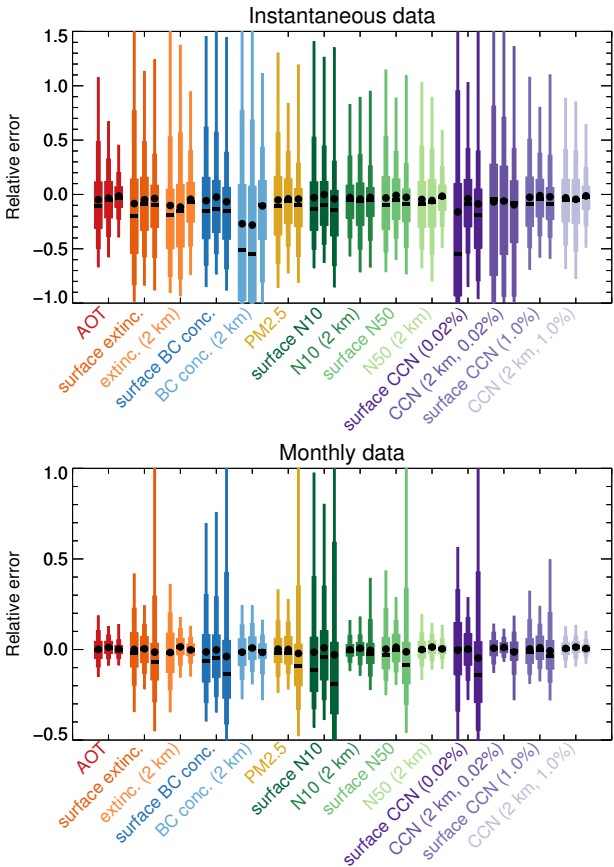

**Figure 8.** Relative spatial sampling errors (for either instantaneous or monthly data, note the different vertical axes) for all WRF-Chem MADE regions (left bar: W-Europe; centre bar: Oklahoma; right bar: Congo). Further explanation in Sec. 3.2.



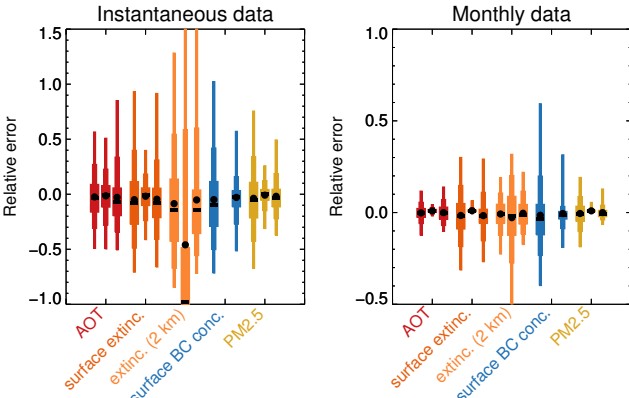

**Figure 9.** Relative spatial sampling errors (for either instantaneous or monthly data, note the different vertical axes) for three regions simulated with mass-bulk schemes (left bar: Europe; middle bar: Ocean; right bar: Japan). Black carbon concentrations over Ocean are zero and so are related spatial sampling errors. Further explanation in Sec. 3.2.

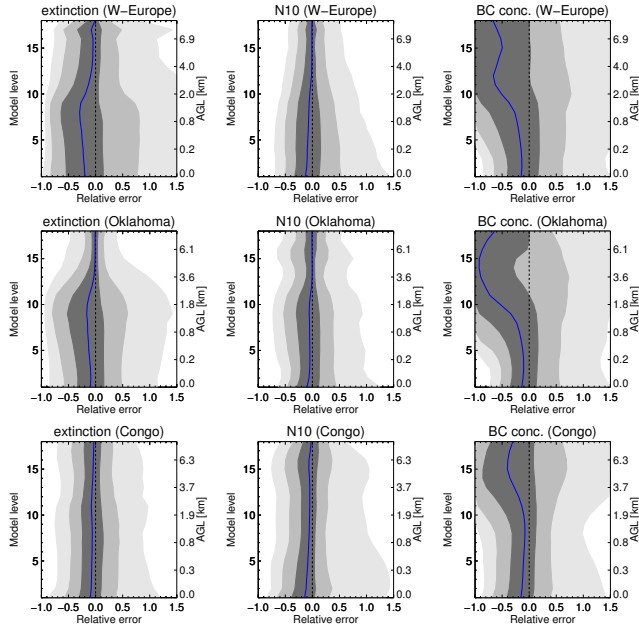

**Figure 10.** Relative spatial sampling error (instantaneous data) as a function of model level (left vertical axis) and altitude above ground level (AGL, right vertical axis) for extinction, N10 and black carbon concentrations. Results for the WRF-Chem MADE simulations. Further explanation in Sec. 3.2.





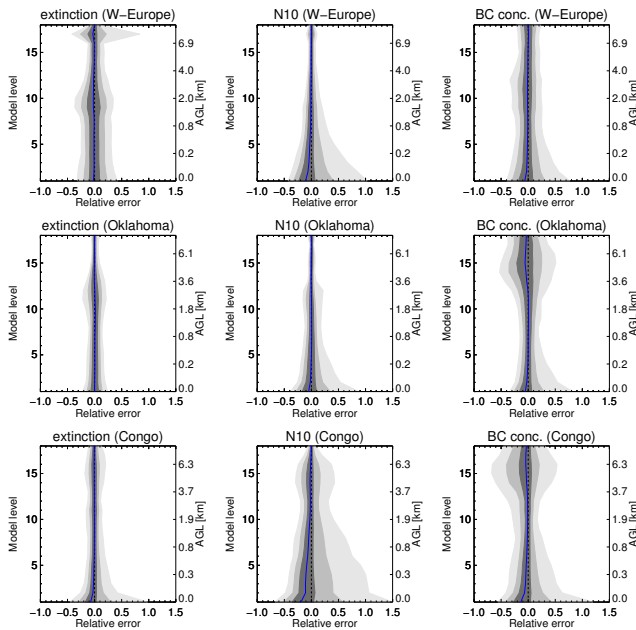

**Figure 11.** Relative spatial sampling error (monthly data) as a function of model level (left vertical axis) and altitude above ground level (AGL, right vertical axis) for extinction, N10 and black carbon concentrations. Results for the WRF-Chem MADE simulations. Further explanation in Sec. 3.2.

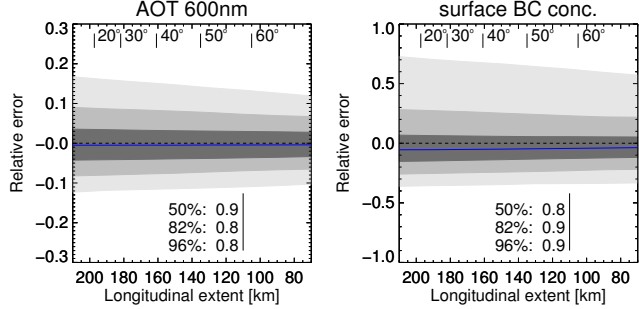

**Figure 12.** Relative spatial sampling errors (monthly data) as a function of longitudinal extent of the grid-box (due to latitude). Near the top horizontal axis, latitudes are given. Near the bottom horizontal axis, the ratios of $\Delta q_{25}, \Delta q_{82}$ and $\Delta q_{96}$ at two different longitudinal extents (110 over 210 km) are given. Results from WRF-Chem MADE over W-Europe. Further explanation in Sec. 3.2.



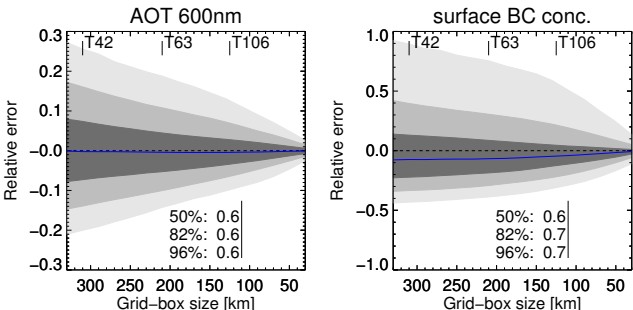

**Figure 13.** Relative spatial sampling errors (monthly data) as a function of grid-box size. Near the top horizontal axis, standard spectral grid sizes are shown. Near the bottom horizontal axis, the ratios of $\Delta q_{25}, \Delta q_{82}$ and $\Delta q_{96}$ at two different grid-box sizes (110 and 210 km) are given. Results from WRF-Chem MADE over W-Europe. Further explanation in Sec. 3.2.

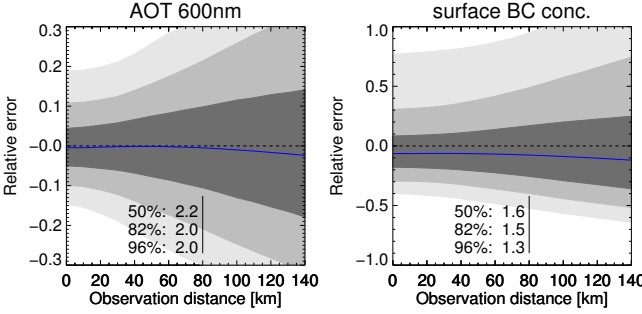

**Figure 14.** Relative spatial sampling error (monthly data) as a function of distance of the observation to the grid-point. Near the bottom horizontal axis, the ratios of $\Delta q_{25}, \Delta q_{82}$ and $\Delta q_{96}$ at a distance of 80 and 0 km are given. Results from WRF-Chem MADE over W-Europe. Further explanation in Sec. 3.2.

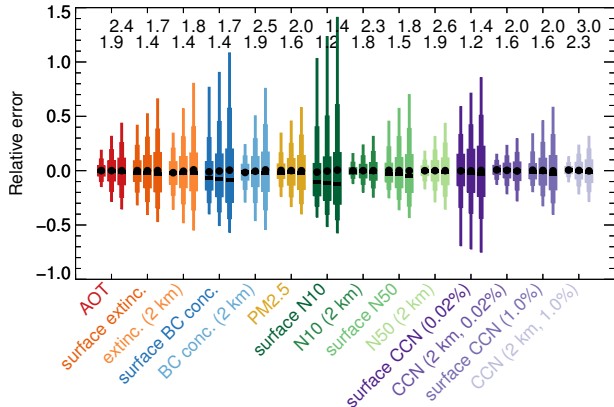

**Figure 15.** Relative spatial sampling error (monthly data) as a function of distance of the observation to the grid-point. The numbers near the top horizontal axis show the increase of $\Delta q_{82}$ at resp. 70 and 100 km relative to 0 km. Results from WRF-Chem MADE over W-Europe. Further explanation in Sec. 3.2.



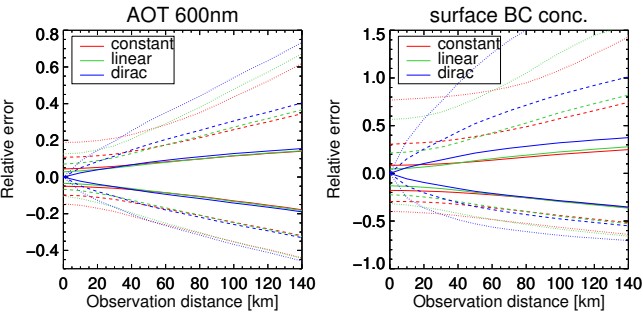

**Figure 16.** Relative spatial sampling error (monthly data) as a function of distance of the observation to the grid-point, for three different weighting functions. Results from WRF-Chem MADE over W-Europe. The usual inter-quantile ranges $\Delta q_{50}$ (solid), $\Delta q_{82}$ (dashed) and $\Delta q_{96}$ (dotted) are shown.





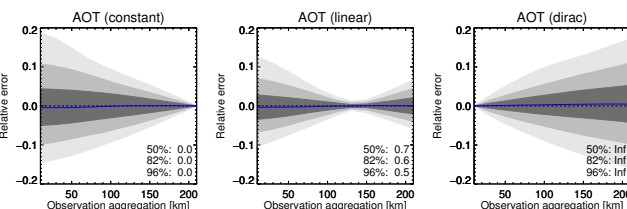

**Figure 17.** Relative spatial sampling error (monthly data) as a function of aggregation extent of the AOT observations, using three different weighting functions. The centre of the aggregated observations is assumed to coincide with the model's grid-points. In the lower right corner, the ratios of $\Delta q_{25}$, $\Delta q_{82}$ and $\Delta q_{96}$ at two different aggregation extents (210 to 0 km) are given. Results from WRF-Chem MADE over W-Europe. Further explanation in Sec. 3.2.

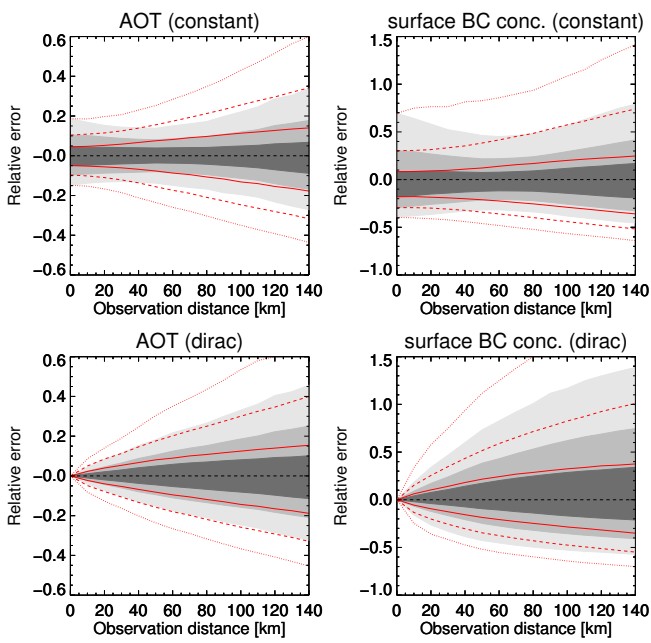

**Figure 18.** Relative spatial sampling error (monthly data) for 4 randomly distributed sites as a function of distance to the grid-point, assuming two different weighting functions. The red lines indicate the errors for a single site (see also Fig. 14). Results from WRF-Chem MADE over W-Europe. Further explanation in Sec. 3.2.





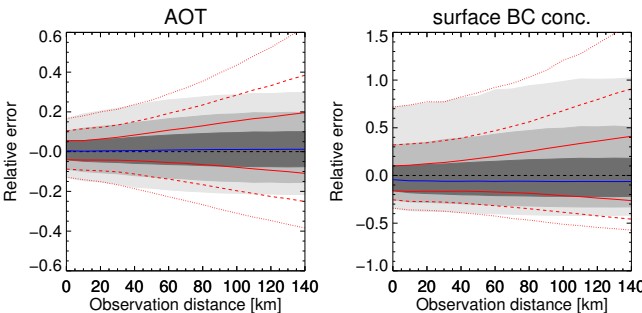

**Figure 19.** Relative spatial sampling error (monthly data) i.c. of linear interpolation of model values to the observation, as a function of distance to the grid-point. The red lines indicate the errors without interpolation (see also Fig. 14). Results from WRF-Chem MADE over W-Europe. Further explanation in Sec. 3.2.

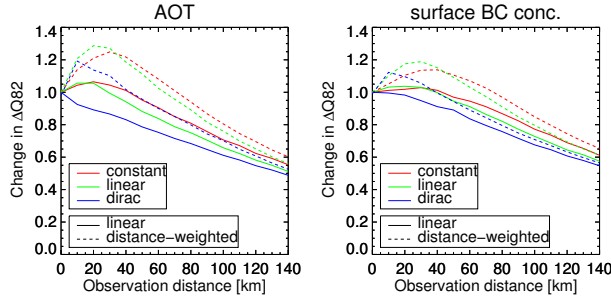

**Figure 20.** Change (relative to Fig. 14) in $\Delta q_{82}$ (for monthly relative sampling errors) due to interpolation, as a function of distance to the grid-point. All three weighting functions and two interpolation methods are considered. Similar graphs for $\Delta q_{50}$ and $\Delta q_{96}$ can be shown.

**Table 1.** Simulations analysed in this study

| region | size [km$^2$] | period | model | scheme | comments |
|---|---|---|---|---|---|
| W-Europe | $1280 \times 1280$ | May 2008 | WRF-Chem | MADE | 2-moments modal |
| Oklahoma | $1190 \times 1190$ | March 2007 | WRF-Chem | MADE | 2-moments modal |
| Congo | $2090 \times 2090$ | March 2007 | WRF-Chem | MADE | 2-moments modal |
| Ocean | $1270 \times 1270$ | March 2007 | WRF-Chem | GOCART | mass bulk |
| Europe | $4000 \times 3100$ | January - June 2008 | EMEP | | mass bulk |
| Japan | $1500 \times 1250$ | August 2007 | NICAM | SPRINTARS | mass bulk |





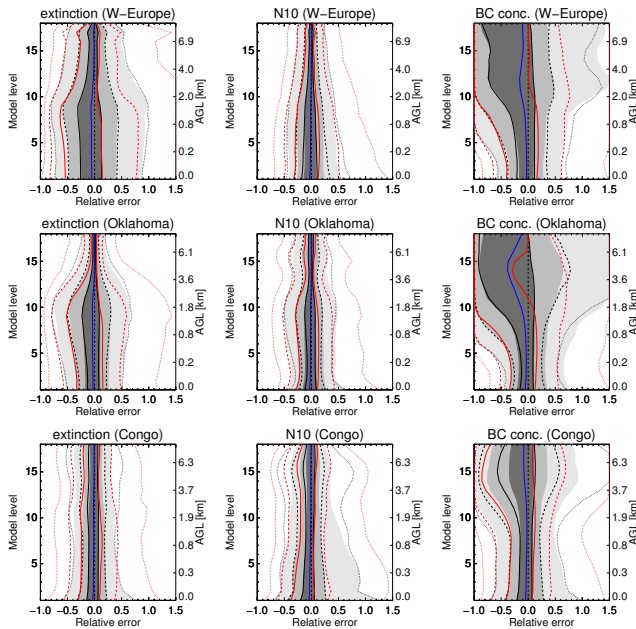

**Figure 21.** Relative spatial sampling error (for measurements during horizontal legs of a flight campaign) as a function of model level (left vertical axis) and altitude above ground level (AGL, right vertical axis) for extinction, N10 and black carbon concentrations. The grey shaded error ranges are for North-South flights. Similar error ranges for East-West flights are shown in black lines. The results of Fig. 10 are also shown in red lines. The usual inter-quantile ranges $\Delta q_{50}$ (solid), $\Delta q_{82}$ (dashed) and $\Delta q_{96}$ (dotted) are shown. Further explanation in Sec. 3.2.

**Table 2.** Simulated observables

|                   | AOT | extinction | $PM_{2.5}$ | BC conc. | N10, N50 | CCN |
|-------------------|:---:|:----------:|:----------:|:--------:|:--------:|:---:|
| WRF-Chem MADE     | ✓   | ✓          | ✓          | ✓        | ✓        | ✓   |
| WRF-Chem GOCART   | ✓   | ✓          | ✓          |          |          |     |
| EMEP              | ✓   | ✓          | ✓          | ✓        |          |     |
| NICAM-SPRINTARS   | ✓   | ✓          | ✓          | ✓        |          |     |