# Peer review of "Will a perfect model agree with perfect observations? The impact of spatial sampling."

_Atmospheric Chemistry and Physics, 2015_

## Referee Comment (RC1) · Anonymous Referee #1 · 17 Feb 2016

In this study, the authors use high-resolution modelling of aerosol distributions to build two datasets, one representative of typical observations and the other of typical global aerosol model simulations. Comparing the two datasets statistically quantifies the errors due to spatial sampling. The authors find that those errors are large, and explore different ways of minimising them, from temporal averaging to model interpolation.

This paper is very interesting and very clearly written. I very much enjoyed reading it. The figures illustrate the discussion very well. I agree with the authors that the lack of previous study of those sampling effects is surprising, given that model/observation comparisons are now a mandatory aspect of most modelling papers and research proposals. That situation strongly suggests that comparisons are often not made carefully, and that observational constraints have often been misleading.
[Figure]

I recommend publication after minor revisions to address the main comments below. The first comment asks for a clear explanation of why errors should increase with distance to the grid-point, a fact that I find difficult to comprehend fully. The second comment requests a more separate discussion of differences of behaviour between observables.

**1  Main comments**

- Sections 4 and 9: I may not be as clever as the average *Atmos. Chem. Phys.* reader, but I have difficulties understanding why errors increase with distance to the grid-point of the 210x210 box. If I understand section 4 and Figure 2 correctly, if $w = 1$, all observations (the 10x10 boxes) within a model 210x210 gridbox are compared to the same value. That value is the average of all observations in the 210x210 gridbox, as calculated by Equation 2. So it should not matter where the 10x10 box is within the 210x210 gridbox, since the value being compared against is always the same. Where do I go wrong here?

- The difference in behaviour between observables is fascinating. Looking at figures 2 and 3, one does not see an essential difference between AOT and BC that could explain the different error statistics (Figure 6) and different responses to temporal sampling (Figure 5, page 7 line 15). Yet they differ. The authors offer hints at possible causes throughout the paper, especially in section 6 where they discuss the narrowness of BC plumes. I recommend adding a more self-contained discussion in the conclusion. That discussion could also be more quantitative. In data assimilation, where they encounter very similar problems, they characterise distributions with correlation length scales. The results of the present paper suggest that AOT, for example, has a longer correlation length scale than BC concentrations, although this is not obvious from looking at Figure 2. Collins *et al.* (2001) use a correlation length scale of 200 km for AOT, which sounds large compared to what the authors imply here. Correlation length scale would also inform the model/observation comparison strategy, with distributions with shorter correlation length scales requiring greater caution and a more adapted distribution of observations.

**2 Other comments**

- Page 4, lines 1–2: SPRINTARS can diagnose number concentrations, but that facility was not used in this study. Is that correct?

- Page 4, lines 12–20: The authors seem to worry about the impact of hygroscopic growth on number concentrations, but I do not understand what the problem is in the context of the study. Can that point be clarified?

- Page 21, caption of Figure 2: The meaning of "at 10 days, 00 hours" is unclear. I suggest "at T+10 days".

- Page 11, section 10.1: Even if I fully understood the reason why errors increase with distance to the grid point, wouldn't that fact be an artifact of the methods used, where model data are regridded versions of observations? In the real world, the two are independent, so distance to gridpoint might be less relevant, undermining the strategy of using only observations close to the gridpoint.

- Page 11, line 30: The distance at which errors are zero in the case of a linear weighting function looks to be two thirds of the gridbox size. Is that expected mathematically, or is it a coincidence?

[Figure]

**3 Technical comments**

- Page 3, line 20: Repeated word "from"
- Page 4, lines 18–19: Reference is wrongly formatted.
- Page 23, Figure 5: It would help if the blue line were thicker.
- Page 7, line 32: Typo: "quite a bit"
- Page 8, line 22: large -> larger
- Page 12, line 16: Typo: "a more localised weighting function"

**4 References**

Collins, W.D., P.J. Rasch, B.E. Eaton, B.V. Khattatov, and J.-F. Lamarque: Simulating aerosols using a chemical transport model with assimilation of satellite aerosol retrievals: Methodology for INDOEX. *J. Geophys. Res.*, 106, 7313–7336, 2001.

---

## Referee Comment (RC2) · Anonymous Referee #2 · 22 Feb 2016

The manuscript analyzes the heterogeneity of aerosol properties within 200 x 200 km2 model grid boxes, using finer-scale simulations for six sets of region and time period. The goal is to reveal the impact of the spatial heterogeneity when combining global models with satellite observations. The main conclusion is that, while the impact depends on matching strategy, it is generally greater than the errors associated with satellite observations. This statement is commonly believed, but has rarely been demonstrated to my knowledge. I recommend publication after the authors address the following minor comments.

Some statements about ground-based observations need to be clarified. In Page 1, Line 5 "the field-of-view of ground-sites" is implied to be 10 km. In Page 2 Line 12 the AERONET is said to sample "no more than 5 km". In fact, most of current ground-based aerosol measurements sample significantly shorter lengths over their native integration

time (between ~1s and minutes). The lengths are some centimeters or meters, depending chiefly on horizontal wind speed and secondarily on instrument flow rate and sunphotometer light collector width. Ground-based measurements represent a distance comparable to 10 km only if integrated over tens of minutes or a couple of hours. Perhaps the authors have an integration time of one hour in mind for the ground-based observations, as this is the temporal resolution of the simulations. This is very different from the native integration time and should be noted.

A few words on the expression of differences and errors would be nice. The expression (observation-model)/model (eq 4) produces numbers widely different from (model-observation)/observation (e.g., +160% vs -62%). The latter expression encounters division by zero when the observation is zero, but is nonetheless fairly commonly used. I would recommend stating explicitly that the present study treats the model as the reference against which the "observation" is evaluated, and not the other way around.

Page 3. Line 20. Remove one of the two "from".

Page 3. Line 28. Replace "main island Japan" with "the largest island of Japan". Japan has three more main islands.

Page 4. Line 10. Move the first parenthesis to immediately before 2013.

Page 4. Line 19. Move the first parenthesis to immediately before Seinfeld.

Page 7. Line 20. I do not see the exception in Figure 6.

Page 7. Line 29. "wet and dry and wet deposition" should read "wet and dry deposition" or "dry and wet deposition".

Page 8. Line 17. Remove "e.g."

Page 8. Line 26. I would think short life-time works to increase spatial heterogeneity, not decrease.

Page 10. Line 14. "Sofar" should read "So far". Also Line 27.

[Figure]

Page 11. Line 25. Remove the hyphen from "More-over".

Page 13. Line 21. Replace no with not.

---

## Author Comment (AC1) · 25 Apr 2016

We thank our two reviewers for carefully reading our manuscript and supplying us with useful comments and constructive criticism.

**Response to reviewer 1**

**Main comments**

*Sections 4 and 9: I may not be as clever as the average Atmos. Chem. Phys. reader, but I have difficulties understanding why errors increase with distance to the grid-point of the 210x210 box. If I understand section 4 and Figure 2 correctly, if w = 1, all observations (the 10x10 boxes) within a model 210x210 gridbox are compared to the same value. That value is the average of all observations in the 210x210 gridbox, as calculated by Equation 2. So it should not matter where the 10x10 box is within the 210x210 gridbox, since the value being compared against is always the same. Where do I go wrong here?*

The reviewer's understanding of our analysis procedure is correct. If the values in the different 10x10 boxes where just random numbers independently drawn from the same distribution, the sampling errors should not depend on distance. However, the observations are not drawn from the same distributions nor are they independent. The true distribution (defined by a multi-year time-series) will depend on the location of the observation relative to sources, with respect to the atmospheric flow. Clearly this will differ for each observation within a 210x210 box. For the same reason, nearby observations will tend to be correlated; both our simulated aerosol and observed aerosol (Anderson et al JAS 2003) exhibit spatial correlations over distances of 10-100 km. Consequently, an observation in the center of a 210x210 box should be strongly correlated with that 210x210 average. But a 10x10 box in the top right corner may only be strongly correlated with the top right quadrant of the 210x210 box but not its bottom-left quadrant. A simpler and less accurate way to put it would be to say that the region for which the observation is representative and the grid-box itself overlap less and less as the distance increases. This hopefully explains the increase of errors with distance. We have added more explanation to Sect 9.

*The difference in behaviour between observables is fascinating. Looking at figures 2 and 3, one does not see an essential difference between AOT and BC that could explain the different error statistics (Figure 6) and different responses to temporal sampling (Figure 5, page 7 line 15). Yet they differ. The authors offer hints at possible causes throughout the paper, especially in section 6 where they discuss the narrowness of BC plumes. I recommend adding a more selfcontained discussion in the conclusion. That discussion could also be more quantitative. In data assimilation, where they encounter very similar problems, they characterise distributions with correlation length scales. The results of the present paper suggest that AOT, for example, has a longer correlation length scale than BC concentrations, although this is not obvious from looking at Figure 2. Collins et al. (2001) use a correlation length scale of 200 km for AOT, which sounds large compared to what the authors imply here. Correlation length scale would also inform the model/observation comparison strategy, with distributions with shorter correlation length scales requiring greater caution and a*

*more adapted distribution of observations.*

We were also surprised by the large difference in behavior. Static graphs like the ones we show in the paper do little to explain this. We have been looking at movies (available as a video supplement) of how the different fields evolve and that is far more instructive. It becomes obvious that bc and number densities in the lowest model layer 'stick' very closely to their sources (they either are deposited or elevated to higher layers). As a result, correlation length scales are very short and a single observation not very representative of the larger grid-box. In contrast, AOT is much better mixed horizontally (plus: vertical transport does not affect it) and shows longer correlation length-scales. We hesitate to explore this more fully in the current paper: it is big enough already and clearly many aspects go into this (it is a bit like asking why different models give different results: a valid question but with no easy answer). Also, we plan to return to this issue in a follow-up paper.

**Other comments**

*-Page 4, lines 1–2: SPRINTARS can diagnose number concentrations, but that facility was not used in this study. Is that correct?*

Yes, that is correct. As SPRINTARS *diagnoses* number densities (instead of calculating them prognostically) we decided there was not much to be gained from an analysis that would essentially yield the same results as say mass concentrations (e.g. pm25). Same for GOCART (WRF-Chem) over ocean.

*– Page 4, lines 12–20: The authors seem to worry about the impact of hygroscopic growth on number concentrations, but I do not understand what the problem is in the context of the study. Can that point be clarified?*

We discuss here how well WRF-Chem is able to simulate properties *as they might be observed*. Many particle counters first dry the aerosol, then filter by size. We only use standard WRF-Chem output which provides us with number densities in each of its three modes *at ambient conditions*. Backing out number densities at dry conditions requires information on aerosol wet-growth which was not readily available (not in the least because of the complex ammonia & nitric-acid & sulfuric-acid & water equilibrium used in WRF-Chem). We have tried to explain this more clearly in the paper.

*– Page 21, caption of Figure 2: The meaning of "at 10 days, 00 hours" is unclear. I suggest "at T+10 days".*

Agreed and changed.

*– Page 11, section 10.1: Even if I fully understood the reason why errors increase with distance to the grid point, wouldn't that fact be an artifact of the methods used, where model data are regridded versions of observations?*

*In the real world, the two are independent, so distance to gridpoint might be
less relevant, undermining the strategy of using only observations close to
the gridpoint.*

For reasons explained before, we consider this effect (increasing error with distance) to be real and directly related to the spatio-temporal structure of the aerosol fields. Obviously, our models may be over- or underestimating the spatio-temporal correlations in these fields.

*– Page 11, line 30: The distance at which errors are zero in the case of a
linear weighting function looks to be two thirds of the gridbox size. Is that
expected mathematically, or is it a coincidence?*

The errors do not become zero, merely small. Intuitively, we can expect this: the linear weighting function effectively defines a smaller grid-box (smaller than 210x210 but much larger than 10x10) for which the global model data is representative. If our observational aggregate approaches this size, errors are likely to be smallest. Mathematically, it will depend on the spatio-temporal correlations in the field itself. In practice, we do not see a big difference in this length scale amongst our six simulations.

**Technical comments**

*– Page 3, line 20: Repeated word "from"*

Corrected

*– Page 4, lines 18–19: Reference is wrongly formatted.*

Corrected

*– Page 23, Figure 5: It would help if the blue line were thicker.*

Corrected

*– Page 7, line 32: Typo: "quite a bit"*

Corrected

*– Page 8, line 22: large -> larger*

Corrected

*– Page 12, line 16: Typo: "a more localised weighting function"*

Corrected

---

## Author Comment (AC2) · 25 Apr 2016

We thank our two reviewers for carefully reading our manuscript and supplying us with useful comments and constructive criticism.

**Response to reviewer 2**

*In Page 1,*
*Line 5 "the field-of-view of ground-sites" is implied to be 10 km. In Page 2 Line 12 the AERONET is said to sample "no more than 5 km". In fact, most of current ground-based aerosol measurements sample significantly shorter lengths over their native integration time (between _1s and minutes). The lengths are some centimeters or meters, depending chiefly on horizontal wind speed and secondarily on instrument flow rate and sunphotometer light collector width. Ground-based measurements represent a distance comparable to 10 km only if integrated over tens of minutes or a couple of hours. Perhaps the authors have an integration time of one hour in mind for the ground-based observations, as this is the temporal resolution of the simulations. This is very different from the native integration time and should be noted.*

The reviewer is correct that the 10 km resolution of our models does not do full justice to the scales at which some observations are made. This is a limitation in our analysis and was discussed in our summary. We point out that, like many atmospheric properties, aerosol shows less variation on short length-scales than on larger length scales (power spectra of aerosol distributions in space or time show a typical power-law behavior). Consequently, we suspect that variability over 10 km will not substantially alter our conclusions (although our results may somewhat underestimate sampling errors).

*A few words on the expression of differences and errors would be nice. The expression(observation-model)/model (eq 4) produces numbers widely different from (modelobservation)/observation (e.g., +160% vs -62%). The latter expression encounters division by zero when the observation is zero, but is nonetheless fairly commonly used. I would recommend stating explicitly that the present study treats the model as the reference against which the "observation" is evaluated, and not the other way around.*

For the reason given by the reviewer, we chose to use model data (210x210 box average) as the reference. This was explained in Sect. 3 but we now repeat it in the Summary and Introduction as well. We've also discussed this in more detail in Sect. 3.

*Page 3. Line 20. Remove one of the two "from".*

Corrected

*Page 3. Line 28. Replace "main island Japan" with "the largest island of Japan". Japan has three more main islands.*

While the reviewer is correct, the island we refer to is called Honshu, which means 'main island'. It would be identified as the main island by most Japanese. In the interest of non-Japanese readers we have changed the text to 'largest island'.

*Page 4. Line 10. Move the first parenthesis to immediately before 2013.*

Corrected

*Page 4. Line 19. Move the first parenthesis to immediately before Seinfeld.*

Corrected

*Page 7. Line 20. I do not see the exception in Figure 6.*

The bars to compare are the two orange bars in the left plot. The left bar is WRF-Chem, the right bar is EMEP. They are quite similar, certainly compared to the other bars in these plots. We have removed this line to prevent confusion.

*Page 7. Line 29. "wet and dry and wet deposition" should read "wet and dry deposition" or "dry and wet deposition".*

Corrected

*Page 8. Line 17. Remove "e.g."*

Corrected

*Page 8. Line 26. I would think short life-time works to increase spatial heterogeneity, not decrease.*

That is a good point. What we see in this simulation is low spatial but high temporal variation over Ocean. Presumably that is due to spatial correlations in rapidly varying windspeeds. In the current paper, only spatial sampling is considered and hence sampling errors over Ocean are small. In a paper that we are currently working on, also temporal sampling is considered which substantially increases errors over Ocean. We have corrected the text and removed reference to short life-times.

*Page 10. Line 14. "Sofar" should read "So far". Also Line 27.*

Corrected.

*Page 11. Line 25. Remove the hyphen from "More-over".*

Corrected

*Page 13. Line 21. Replace no with not.*

Corrected.